# Exploring Forensic Dental Identification with Deep Learning

**Yuan Liang**[1,2], **Weikun Han**[1], **Liang Qiu**[1], **Chen Wu**[1], **Yiting Shao**[1], **Kun Wang**[1]*, **Lei He**[1]*
[1]University of California, Los Angeles
[2]Topaz Labs
liangyuandg@ucla.edu

## Abstract

Dental forensic identification targets to identify persons with dental traces. The task is vital for the investigation of criminal scenes and mass disasters because of the resistance of dental structures and the wide-existence of dental imaging. However, no widely accepted automated solution is available for this labour-costly task. In this work, we pioneer to study deep learning for dental forensic identification based on panoramic radiographs. We construct a comprehensive benchmark with various dental variations that can adequately reflect the difficulties of the task. By considering the task's unique challenges, we propose FoID, a deep learning method featured by: (*i*) clinical-inspired attention localization, (*ii*) domain-specific augmentations that enable instance discriminative learning, and (*iii*) transformer-based self-attention mechanism that dynamically reasons the relative importance of attentions. We show that FoID can outperform traditional approaches by at least **22.98%** in terms of Rank-1 accuracy, and outperform strong CNN baselines by at least **10.50%** in terms of mean Average Precision (mAP). Moreover, extensive ablation studies verify the effectiveness of each building blocks of FoID. Our work can be a first step towards the automated system for forensic identification among large-scale multi-site databases. Also, the proposed techniques, *e.g.*, self-attention mechanism, can also be meaningful for other identification tasks, *e.g.*, pedestrian re-identification. Related data and codes can be found at https://github.com/liangyuandg/FoID.

## 1 Introduction

Forensic identification targets to identify living or deceased persons by analyzing their trace evidences. The identification with dental data is particularly vital for the investigation of criminal scenes, accidents and mass disasters for at least two reasons [30, 27]: (*i*) dental patterns can be highly identifying while their traces are widely archived than other methodologies *e.g.,* DNA profiling [4, 45, 59]. For instance, around 1.4 billion dental X-rays were performed in 2019 in the U.S.[2]; (*ii*) dental structures are more resistant to damages than other body tissues including bones [7, 46]. In the current practice, forensic dental examinations are mostly done by the visual comparison of radiographs [46, 45] between a target entity (person) and those from databases. Due to the lack of widely accepted automated solutions, such examination cannot scale up to large databases, and its results are vulnerable to oversights and/or mistakes [19, 39]. This work pioneers to study deep learning for forensic identification with dental panoramic radiographs — one of the most common dental traces nowadays [59].

---

*Corresponding authors
[2]https://idataresearch.com/product/dental-imaging-market/

35th Conference on Neural Information Processing Systems (NeurIPS 2021).

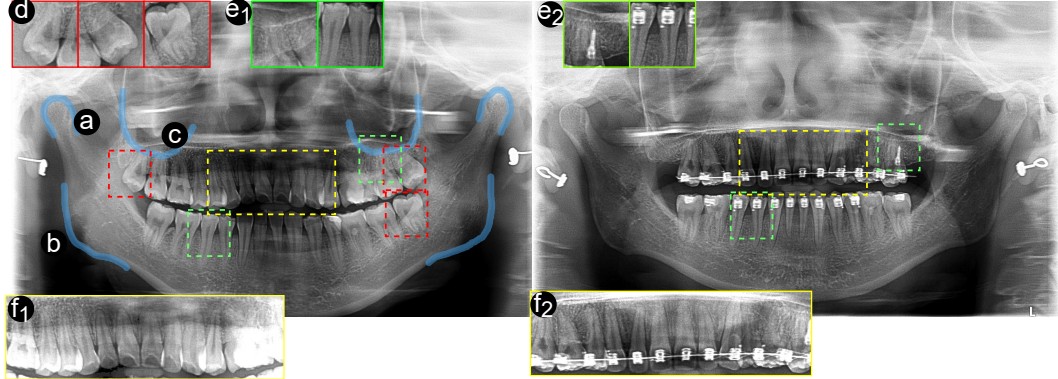

Figure 1: A challenging case of dental identification with two dental radiographs captured from a same person. The case exists large anatomical variations including tooth reduction (d), the addition of braces and implants (e1, e2), and the change of teeth alignment (f1, f2). In the forensic practice, teeth, condyle (a), angle of mandible (b), and maxillary sinus (c) are the key anatomies for dentists.

There have been a few attempts for forensic dental identification, most of which take the approaches of matching handcrafted feature [1, 78, 9]. According to our best knowledge, only one work utilizes off-the-shelf convolutional neural networks (CNNs) for learning discriminative representations of a dental radiograph for identification [39]. However, the work is preliminary since it is based on a *small-scale* dataset with both *temporal* and *heterogeneous* limitations — it is different from the applied situation where significant intra-entity variations in dental structures can exist because of external factors, *e.g.*, interventions, and internal factors, *e.g.*, decaying, over long scanning intervals [46, 1]. Moreover, no architecture exploration was made for this unique identification task.

We performed a formative study with two board-certified dentists, and outlined the distinct challenges of forensic dental identification: (*C1*) the lack of paired dental radiographs for metric learning; (*C2*) sparseness of identification information in high-resolution radiographs; and (*C3*) dental variations can be heterogeneous but useful as *inclusion* or *exclusion* criteria for identifying. To study the task, we construct a comprehensive benchmark: it involves 583 persons, and comes with a challenging testing set spanning over $21.0\pm11.5$ months of scanning intervals, covering a wide range of dental variations including orthodontics, tooth loss, and implanting/filling. Moreover, we propose the first optimized deep learning **Fo**rensic **ID**entification solution, named **FoID**. In specific, to take the advantage of overwhelmingly unpaired radiographs (*C1*), we develop a set of domain-specific augmentations (DSA) for effective self-supervised metric learning. To focus on meaningful areas, FoID incorporates hard attention extraction and alignment guided by semantic segmentation and anatomical registration (*C2*). Moreover, FoID introduces a novel transformer-based self-attention mechanism, as inspired by dentist's practice, to dynamically reason the optimal aggregation of partial attentions for representation (*C3*). Overall, the main contributions of our work are three folds:

- We give the first in-depth study of deep learning for dental forensic identification. The formative study with dentists outlines the unique challenges and offers insights into solutions.
- We construct a comprehensive dental forensic benchmark, which covers a wide range of dental variations to reflect the realistic scenario of the task.
- We present the first deep learning solution FoID, which achieves an mAP of 59.62%, and largely outperforms the existing dental identification models and strong CNN baselines. Ablation studies show that the proposed DSA enables effective representation learning from unpaired data, boosting models by up to 12.33% in terms of mAP; Meanwhile, the proposed transformer-based self-attention outperforms state-of-the-art attention models by at least 4.82% in terms of mAP.

## 2 Formative Study

**Challenge 1: paired scan instances are scarce**   With the high radiation of dental imaging, overwhelming cases from dental clinics contain only one scan instance per person. As such, it poses as a challenge for metric learning without positive instance pairs. To tackle it, we develop a set of domain-specific augmentations (DSA) by working with dentists to simulate potential dental

anatomical variations. Different views of one entity are generated on the fly, which enables effective intra-/inter-reasoning in a self-supervised manner.

**Challenge 2: identification information is sparse**   Forensic dentists mostly focus on several key anatomies on panoramic imaging for identification, since others are either non-discriminative or not well approximated on radiographs. As such, models should focus on such regions for image embedding to avoid over-fitting. Accordingly, as shown in Figure 1, our method such incorporate domain knowledge by utilizing a hard attention mechanism on teeth, condyle (Figure 1(a)), angle of mandible (Figure 1(b)), and maxillary sinus (Figure 1(c)). A combination of semantic segmentation and registration is applied for efficient key-point localization.

**Challenge 3: dental variations are heterogeneous but can be useful**   There can exist both local and global anatomical changes over two scanning. As demonstrated by a typical case in Figure 1, two scans of a same entity can see anatomy loss (Figure 1(d)), artifact implanting (Figure 1(e1, e2)), structural changes (Figure 1(f1, f2)). In the forensic practice, dentists commonly look through all key anatomies in a scan, determine most discriminative features among them according to experience, and then apply those as indexes for matching. Our method follows the philosophy by introducing a transformer-based self-attention module, which dynamically reasons the relative importance of attentions among a set of them, for the effective representation learning of a radiograph.

## 3   Related Works

**Person identification with deep learning**   Person identification tasks such as face recognition [56, 52, 43, 47, 8] and pedestrian re-identification [13, 50, 63, 60] have been widely studied by the computer vision community. Current solutions either optimize a discriminant distance metric that takes a pair of instances [69, 79, 61, 3] as input, or learn an identity-sensitive encoder that maps an input instance to representations [74, 50, 77, 57]. Our method follows the later approach for its efficiency of inference within large-scale galleries.

*Attention mechanism*  has been widely explored for pedestrian re-identification to enforce models to focus on key body parts, such that representations are robust against background clutter, occlusion, and unconstrained human poses in surveillance images. For example, existing works [51, 40, 11, 54, 62, 63, 33, 66, 48, 74] extract part-based features with either soft or hard attentions, guided by human pose estimations, and align/aggregate them for the final representation. Works [35, 53, 60] further propose to mine useful partial attentions from the end-to-end training. Our work follows the approach by introducing an attention mechanism, but differs in two ways. First, the key areas are partially localized with non-rigid image registration [5], which effectively reduces the cost of manual annotations. Second, more importantly, we propose a novel self-attention module to dynamically reason the relative importance of each attention for the final representation by learning from training cases. Such self-attention is inspired by the practice of experts, where unique features of dental structures is searched, and used as matching criteria.

*Unsupervised person identification*  has also been recently discussed in pedestrian re-identification in order to train models with the widely available unlabeled surveillance videos. The current solutions can be categorized into clustering and fine-tuning [21, 75, 17, 20, 72], negative instance pair mining [70, 37], and image translation [6, 64, 77, 65]. However, the challenge of forensic dental identification is different from the above scenario: within a dental image database, each entity (person) has only one scan instance (scan), and the method should learn representations from such set of instances. As such, we take the instance discriminative approach [10, 44, 21], and generate different views of the same entity with the proposed domain-specific augmentations (DSA) for the contrastive intra-reasoning.

**Transformer for image retrieval**   Transformer has been recently applied to computer vision tasks [15, 71, 73]. Since our task is image retrieval in nature, we briefly review the related transformer works: Gkelios *et al.* [22] and El-Nouby *et al.* [16] directly replace CNNs with pure transformer models for image embedding; He *et al.* [25] further incorporates encoding of meta information, (*e.g.,* camera viewpoints), into the sequence of inputs; Yang *et al.* [67] utilizes query and gallery images as transformer's query and key inputs respectively for modeling spatial attention between them. Different from the above works, we propose to utilize a transformer for aggregating a set of model attentions: the self-attention characteristic of transformer enables the dynamic reasoning of

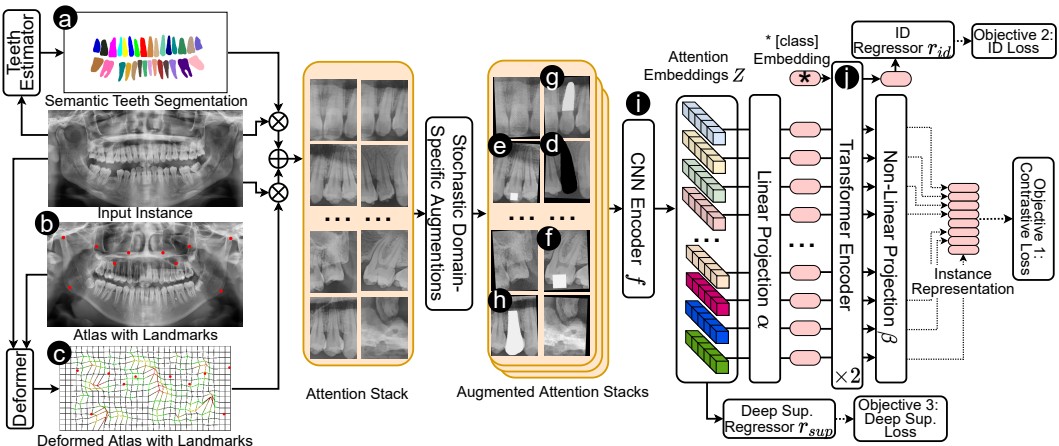

Figure 2: Overall architecture of FoID, the key components of which consist of attention localization and alignment, domain-specific augmentations, and feature aggregation with self-attention. The objective function includes three types of losses that are complementary for representation learning.

attentions' importance for embedding by comparison, and its permutation-invariance [32] can relieve the attention misalignment caused by the imperfect key-point localization.

## 4   Methodologies

We present FoID by first describing its building components and then the multi-task learning strategy. As shown in Figure 2, FoID maps each scan instance into an embedding. Given a query instance, we calculate the similarities between the query and all the gallery instances in the embedding space and return the ranked retrieval list.

### 4.1   Attention Localization & Alignment

Inspired by the dentists' practice, FoID employs a hard attention mechanism to focus on (*i*) teeth and (*ii*) landmarks of condyle, angle of mandible, and maxillary sinus to avoid over-fit to unrelated regions. For teeth localization, FoID formulates it as semantic segmentation, and applies a mask-RCNN [23] pre-trained on the existing panoramic segmentation dataset of UFBA_UESC [29]. Given an input instance $I$, a teeth segmentation map $T^{H \times W \times 32}$ is estimated (Figure 2(a)), where $H$ and $W$ are instance height and width, and a channel size is 32 for the number of teeth categories of an adult. The raw mask-RCNN outputs are filtered for the largest connected mask with a minimal size threshold for each tooth to denoise. For landmark localization, since the landmarks are largely separated and establish no accurate boundary, FoID employs a registration-based method to efficiently detect them from an atlas, with one pair of radiograph and its manual annotation with $M=6$ landmarks manually created (Figure 2(b)). For each input instance, the atlas is aligned to the instance with Syn registration [5], with the annotation deformed accordingly for the landmark estimation (Figure 2(c)). Based on the teeth and landmark localization, the input is cropped to attention patches with a constant spatial dimension of $h \times w$, and then sequentially piled as the attention stack $S^{h \times w \times 38}$ for the representation representation. In the case of missing anatomies, zero-valued patches are correspondingly padded. **Appendix** includes detailed results on teeth segmentation, landmark localization, and the visualization of the annotated atlas.

### 4.2   Contrastive Learning with DSA

In order to build useful representations from overwhelming unpaired instances, FoID adapts a self-supervised approach with the proposed domain-specific augmentations (DSA). In specific, a set of structure-aware and forensic-realistic augmentations is stochastically applied to an instance to generate different views, which are in turn used for intra-reasoning. In the practice, given a mini-batch of stacked attentions from $N$ entities $B = \{S_n\}_{n=1}^{N}$, $K$ times of DSA are repeatedly applied to $B$. Denote $S_n^{(i)}$ as the $i$-th sample from augmented view of $S_n$: an instance group $\{S_n^{(i)}, S_n^{(j)}\}$ is

taken as positive pair, while $\{S_n^{(i)}, S_{\backslash n}^{(j)}\}$ is taken as a negative pair, where $\backslash n$ can be any index from $\{1, 2, ..., N\}\backslash\{n\}$. The learning strategy does not relay on any paired instances or identification annotation, but can be easily extended when such data is available by including different augmented inputs belonging to the same identification as positive pairs.

To achieve effective representation learning, augmentations that cover the potential anatomical variations as in clinics is the key. As showcased in Figure 2, the proposed DSA consists fours types of augmentations inspired by forensic observations, and is independently applied to each attention patch on the fly. (*i*) Random tooth reduction that aims to simulate tooth loss caused by injury or inventions (*e.g.*, Figure 2(e)). (*ii*) Random artifact addition that adds patches of flexible shapes to crown (Figure 2(e,f)) or tooth regions (Figure 2(g,h)). Such augmentation aims to simulate the common artifacts of dental filling, implant, and brace. (*iii*) Random rigid patch transform within a range of angles and displacements. The augmentation not only enforces view comparison as in the standard instance discrimination learning [10, 44], but also simulates global teeth arrangement changes (*e.g.,* caused by orthodontics), since the transform is performed for patches independently. (*iv*) Random color disturbance of contrast shifting and gaussian noise within designed ranges for simulating different scanning setup and machine noises. More details about the implementation of DSA are included in **Appendix**.

### 4.3  Self-Attention for Feature Aggregation

To build the representation of an instance from its attention stack, FoID consists a CNN encoder $f$ (Figure 2(i)) to embed each attention patch, followed by a transformer-based self-attention module (Figure 2(j)) for aggregating the attention embeddings. The encoder $f$ targets to extract salient features for an input patch with the existence of possible anatomical variations, and it can be adapted from any convolutional encoders. The output attention embeddings from the CNN encoder can be denoted as $Z^{38 \times C}$, where $C$ is the embedding length.

To aggregate the generated multiple embeddings of attentions, we propose to utilize a transformer to dynamically reason the importance of each attention among the set of them: dental structures that fall into the general distribution of populations should take less attentions, while the ones that are more identifying should take more attentions. More importantly, such importance of attentions should be relative based on the experience of forensic dentists: the more discriminating anatomies in an instance are more often selected as index for matching. This differs from the existing pedestrian re-identification works [11, 33, 74], where the attention to a body parts is modeled without conditioning on its comparison among all body parts. As shown in Figure 2, the sequence of attentions $Z^{38 \times C} = [z_1^C, z_2^C, ..., z_{38}^C]$ is first projected into a constant token dimension $C'$ with a linear encoder $\alpha$, and then is fed into a two-layer BERT encoders [58, 14] for unidirectional context learning. Note that *no* positional information [15] is injected to attention embeddings for two reasons: (*i*) we assume that the semantic category of anatomies sets no prior for reasoning the importance, (*ii*) teeth can be often mis-classified due to their similarity in patterns (see **Appendix** for examples), which can in turn pose as noise when positions are modeled. The feature outputs from the last transformer layer are further encoded into a reduced dimension $C''$ as a regularization with a non-linear encoder $\beta$, and then concatenated as the final embedding $Y = cat(y_1^{C''}, y_2^{C''}, ..., y_{38}^{C''})$ for the instance. Moreover, a learnable $[class]$ token $z_0^{C'}$ is prepended to the transformer's input sequence, the embedding of which at the output $y_0^{C'}$ serves as a complementary instance embedding for the purpose of multi-task learning.

### 4.4  Objective Function & Training

To learn discriminative representations, we adopt three complementary optimization tasks to train FoID: a contrastive loss, an identification loss, and a deep supervision loss.

**Contrastive loss**   Given a pair of instance embeddings $\{Y_i, Y_j\}$, the contrastive loss $L_{contrast}$ is defined as: $L_{contrast} = \mathbb{1}_{l_i = l_j}[m_{pos} - S(Y_i, Y_j)]_+ + \mathbb{1}_{l_i \neq l_j}[S(Y_i, Y_j) - m_{neg}]_+$, where $[x]_+$ equals $max(x, 0)$, and $l_i$ denotes the instance's id category of representation $\{Y_i\}$ among the training set; $m_{pos}$ and $m_{neg}$ are similarity margins for positive and negative pairs; $S(.,.)$ is the pre-defined similarity measure, and cosine distance is used in this work. Online mining is applied to train on challenging positive (views of one instance) and negative (views of different instances) pairs.

Table 1: Demographic and dental variation information of the testing set. Scan interval is the average of largest time interval between any two radiographs of an entity. The numbers of entity presenting the dental variations and their proportions are shown.

| Demographic | | | Dental Variations | | | |
|---|---|---|---|---|---|---|
| Age | Gender | Scan Interval | Orthodontics | Filling/Implant | Brace | Tooth Loss |
| $10\sim55$ | F:23; M:17 | $21.0\pm11.5$ mo. | 25 (62.5%) | 10 (25.0%) | 16 (40.0%) | 18 (45.0%) |

**Identification loss**  An identification loss $L_{id}$ is applied to all samples within a mini-batch for the prediction of ID category from the transformer's $[class]$ embedding output. In specific, $L_{id} = CE(softmax(r_{id}(y_0^{C'})), l)$, where $CE$ denotes cross entropy loss; $y_0^{C'}$ and and $l$ are the sample's transformer's $[class]$ output and id category label, respectively; $r_{id}$ is a regressor that transfer the embedding dimension from $c'$ to the total training id count. The identification loss and contrastive loss are applied on different embeddings and have a diverged objective, which can potentially serve as a regularization for the transformer.

**Deep supervision loss**  The CNN encoder $f$ is expected to extract salient features that are discriminative for identification from a given attention patch. Since the target diverges from the contrastive loss and identification loss that directly optimize on the aggregated embeddings, we incorporates a deep supervision signal for $f$. In specific, we formulate the deep supervision loss $L_{sup}$ as a cross entropy loss: $L_{sup} = CE(softmax(r_{sup}(cat(y_1^C, y_2^C, ..., y_{38}^C))), l)$ , where $r_{sup}$ is a regressor that maps the concatenation of a sample's attention embeddings to a vector, the dimension of which equals the total count of training instances.

The final objective function for training FoID is the unweighted sum of the three types of loss: $L = L_{contrastive} + L_{id} + L_{sup}$. During the training, the whole model can be optimized from end-to-end, while the transformer's non-class embeddings $cat(y_1^{C''}, y_2^{C''}, ..., y_{38}^{C''})$ is taken as the representation of an instance during the inference.

## 5 Benchmark Construction

Similar to the setup of existing image retrieval datasets [34, 76, 36], our benchmark consists of a training set and a testing set. The training set is derived from an existing DNS Panoramic dataset [49], which contains 543 panoramic radiographs and exist common dental variations of tooth reduction and artifact addition. Each of the radiographs is scanned from a separate identity, which resembles the practical situation of dental forensic where there is a lack of paired data for metric learning. There are 271 images that overlap between this training set and the one used for teeth localization as aforementioned in Section 4.1. This setup simulates the potential forensic application: a subset of available data in the imaging repository can possibly be labeled for developing the teeth segmentation model. Moreover, the testing data of identification is unseen to the training of segmentation model, and thus would not artificially inflates the performance during the test.

The testing set contains 87 panoramic radiographs scanned with 40 identities, among which 33 identities hold 2 radiographs and 7 identities hold 3 radiographs. For covering comprehensive dental variations to make it sufficient for reflecting the difficulty of dental forensic, the testing set are selected from both endodontics and orthodontics departments. The method is distinct from previous dental forensic studies that either construct testing cases without controlling data variety [18, 1, 26], or consider only a specific dental variation over a short scanning interval (*i.e.*, single tooth extraction over 64.0 days [39]). Table 1 describes the profile of the testing set, and Figure 1 shows one typical case. All the data has been anonymized. To evaluate a model, we apply an *one*-against-*all* strategy for the efficient use of data. In specific, one instance from the testing set is used as the query, while the union of training set and the rest of testing set is used as the gallery set. The average results among all testing instances is reported as the final accuracy measurement.

Table 2: Comparison to the existing dental forensic identification approaches and strong CNN baselines. AHI [26] is not designed to rank all gallery instances, and thus mAP cannot be measured.

| Methods | Backbone | mAP | Rank-1 | Rank-5 | Rank-10 |
|---|---|---|---|---|---|
| AHI [26] | - | - | 0.2759 | 0.4483 | 0.5287 |
| HIDPR [42] | - | 0.3018 | 0.2759 | 0.4138 | 0.4713 |
| classification | ResNet-34 | 0.4912 | 0.4253 | 0.6437 | 0.7356 |
| contrastive | ResNet-34 | 0.4072 | 0.3218 | 0.5977 | 0.6322 |
| triplet | ResNet-34 | 0.4723 | 0.4253 | 0.6092 | 0.6667 |
| reconstruction | ResNet-34 | 0.3537 | 0.3333 | 0.4943 | 0.6092 |
| classification | Inception-ResNet | 0.3638 | 0.2644 | 0.5287 | 0.5862 |
| contrastive | Inception-ResNet | 0.3854 | 0.3448 | 0.4943 | 0.6092 |
| triplet | Inception-ResNet | 0.3609 | 0.2989 | 0.5172 | 0.6437 |
| reconstruction | Inception-ResNet | 0.2912 | 0.2299 | 0.5172 | 0.5862 |
| **FoID (ours)** | ResNet-34 | 0.5465 | 0.4817 | 0.6897 | 0.7471 |
| **FoID (ours)** | Inception-ResNet | **0.5962** | **0.5057** | **0.7586** | **0.7816** |

# 6 Experimental Results

## 6.1 Implementation Details

We adopt two mostly used CNN architectures in re-identification works, ResNet-34 [24] and Inception-ResNet [55], for the CNN encoder $f$. Both models drop the final fully connected layer, and are initialized by pre-training on the ImageNet [12] and the face recognition dataset LFW by following [47], respectively. The output embedding of both encoders has a 512 channels per input attention patch. By following [15], the single layer linear encoder $\alpha$ maps an attention embedding to 128 channels; The transformer holds a hidden embedding dimension of 128 and a multi-attention head number of 4; Meanwhile, the final encoder $\beta$ consists a fully connected layer and batch normalization, mapping the transform's outputs to 13 channels per attention patch, and thus leading to a dimension of 494 for the final instance embedding. The deep supervision regressor and [*class*] embedding regressor are both implemented with two fully connected layers, the output feature dimensions of which are 256 and the total count of training instances respectively. Batch normalization and ReLU activation are applied in both regressors.

Regarding data pre-processing, all the radiography instances are normalized to a pixel spacing of $0.11mm \times 0.11mm$, and are normalized to a same color space with histogram matching. The attention patch dimensions $h$ and $w$ are set as the minimal embodying size for all anatomies, which are $28.16mm$ and $21.56mm$ respectively. Unless otherwise specified, the batch size is set to 6 and the augmentation repeat is set to 5 per instance for all the training. Unless otherwise specified, Adam optimizer is employed with an initial learning rate of $1e$-4, and is reduced to a half every 30 epochs. All the methods are trained for 100 epochs. For the online hard pair mining, the positive and negative similarity margins are set to 0.6 and 0.4 respectively. All experiments are conducted on servers with three GeForce GTX 1080 Ti GPUs. Cosine similarity is applied to rank the retrievals given a query instance. Rank-k retrieval accuracy ($k \in \{1, 5, 10\}$) and mean average precision (mAP) are used as evaluation metrics.

## 6.2 Comparison to Existing Approaches

We compare our method to both: (*i*) key-point matching based on handcrafted descriptors, and (*ii*) strong CNN baselines. For the key-point matching, it is the most studied approach for forensic dental identification. Two recent methods of AHI [26] and HIDPR [42] are included in comparison, which are based on the matching of Speeded Up Robust Features and tooth-aware appearance and positional similarity, respectively. Both methods have achieved satisfying identification accuracies with panoramic radiographs based on their own in-house datasets. For the CNN approach, few study has investigated it possibly due to the challenge of learning from the overwhelming unpaired data. For fair comparison, we enable the proposed domain-specific augmentations for all the training, and include four most commonly used representation learning methods as the strong baselines, the objective functions of which are id classification loss [56, 77], contrastive loss [52, 57, 13], triplet loss [43, 47, 50], and reconstruction loss [31]. As reported in a recent comparison study on pedestrian re-identification [41], the above baselines can already achieve accuracy comparable to the state-of-the-arts. Note that all the CNN baselines take a whole radiograph as input without attention mechanism, and apply the same backbone architectures of ResNet-34 and Inception-ResNet as FoID.

Table 3: Performance of FoID across ages and genders measured in mAP, and the significance values of the observed difference between the corresponding groups.

| | Groups | mAP | 95% CI | $p$-value |
|---|---|---|---|---|
| Age | <=30 | 0.6322 | 0.5161-0.7483 | 0.2703 |
| | >30 | 0.5161 | 0.3260-0.7062 | |
| Gender | Female | 0.6353 | 0.5033-0.7674 | 0.2969 |
| | Male | 0.5296 | 0.3806-0.6787 | |

According to Table 2, our method significantly outperforms all the existing approaches, achieving up to 59.62% in terms of mAP and 50.57% in terms of rank-1 accuracy. Compared to the key-point matching approach, the improvement shows that the proposed augmentation strategy enables effective representation learning from unpaired data. Compared to CNN baselines, our method surpass by incorporating domain knowledge with attentions — the accuracy boost of at least 10.50% in terms of mAP indicates that the localized attention regions are valid for identification, and can obviously reduce the over-fitting to the non-related areas. Examples that visualize the query images and corresponding FoID's retrievals can be found in **Appendix**.

As shown in Table 3, we also report the performance of our proposed FoID across two age groups (<=30 and >30) and two gender groups (female and male). Overall, according to the paired t-test, no significant difference in the model's performance ($p$-value > 0.1) is observed between the aforementioned groups. Meanwhile, we can see that FoID has a lower mAP on the group of age>30. This can possibly be explained as artifacts, *e.g.*, implants and fillings, could be more often introduced to the identities of this group, which makes the identification task more challenging.

## 6.3 Ablation Studies

Table 4: Ablation studies of our proposed self-attention by comparing with the state-of-the-art attention-based identification methods and common aggregation mechanisms.

| Methods | Backbone | mAP | Rank-1 | Rank-5 | Rank-10 |
|---|---|---|---|---|---|
| PDC [51] | - | 0.4556 | 0.4253 | 0.5632 | 0.6322 |
| PGFA [40] | - | 0.4982 | 0.4598 | 0.5862 | 0.6092 |
| Mancs [60] | - | 0.4723 | 0.4368 | 0.5747 | 0.6092 |
| Avg Pool. | ResNet-34 | 0.3707 | 0.3103 | 0.5057 | 0.5977 |
| Max Pool. | ResNet-34 | 0.2519 | 0.2069 | 0.3793 | 0.4253 |
| FC | ResNet-34 | 0.4000 | 0.2989 | 0.5517 | 0.6667 |
| Att | ResNet-34 | 0.4085 | 0.3563 | 0.5172 | 0.5632 |
| Gated Att | ResNet-34 | 0.4268 | 0.4023 | 0.4828 | 0.5060 |
| **FoID (ours)** | ResNet-34 | **0.5465** | **0.4817** | **0.6897** | **0.7471** |
| Avg Pool. | Inception-ResNet | 0.3824 | 0.2989 | 0.5517 | 0.6092 |
| Max Pool. | Inception-ResNet | 0.2654 | 0.1954 | 0.3793 | 0.5172 |
| FC | Inception-ResNet | 0.3007 | 0.2414 | 0.4138 | 0.5172 |
| Att | Inception-ResNet | 0.3527 | 0.2989 | 0.5172 | 0.5517 |
| Gated Att | Inception-ResNet | 0.3741 | 0.3103 | 0.5287 | 0.6322 |
| **FoID (ours)** | Inception-ResNet | **0.5962** | **0.5057** | **0.7586** | **0.7816** |

Table 5: Ablation studies of the objective function. Our proposed function combining three complementary types of losses leads to the overall highest accuracy as indicated by mAP.

| # | $L_{contrast}$ | $L_{id}$ | $L_{sup}$ | mAP | Rank-1 | Rank-5 | Rank-10 |
|---|---|---|---|---|---|---|---|
| | | ResNet-34 as backbone | | | | | |
| 1 | ✓ | | | 0.4760 | 0.4253 | 0.6303 | 0.6811 |
| 2 | | ✓ | | 0.3196 | 0.2874 | 0.4598 | 0.5434 |
| 3 | ✓ | ✓ | | 0.5394 | 0.4598 | 0.6782 | 0.7356 |
| 4 | ✓ | ✓ | ✓ | **0.5465** | **0.4817** | **0.6897** | **0.7471** |
| | | Inception-ResNet as backbone | | | | | |
| 5 | ✓ | | | 0.4438 | 0.3793 | 0.5747 | 0.6092 |
| 6 | | ✓ | | 0.2808 | 0.1609 | 0.4828 | 0.5517 |
| 7 | ✓ | ✓ | | 0.5394 | 0.4368 | 0.7356 | **0.8046** |
| 8 | ✓ | ✓ | ✓ | **0.5962** | **0.5057** | **0.7586** | 0.7816 |

**Self-attention with transformer** FoID incorporates a novel transformer-based self-attention mechanism to reason the relative importance of attentions for the feature aggregating. In order to verify its effectiveness, we first compare with the state-of-the-art attention mechanisms introduced in pedestrian re-identification: PDC [51], PGFA [40], and Mancs [60]. To adapt those methods to our task, the original human body attention localization is replaced by the dental one as beforehand described in subsection 4.1. For fair comparison, the same DSA and training strategy as FoID are applied. Table 4 shows that FoID can outperform all the methods, exceeding by up to 9.80% and 8.04% in terms of mAP and rank-1 accuracy respectively. Different from the observation in [40, 60], FoID outperforms PGFA and Mancs by applying hard attentions over soft attentions — such mechanism provides (*i*) the accurate definition of key anatomies informed from the formative study, and (*ii*) the improved efficiency of training with patches cropped out of large unrelated regions.

More importantly, the results also prove that modeling self-attentions can potentially lead to better aggregation results. To verify, we ablate the effect of self-attention by replacing the transformer with the existing aggregation modules of average pooling (Avg Pool.), max pooling (Max Pool.), fully connected layers (FC), weighted attention (Att) [28], and gated attention (Gated att) [68, 28],

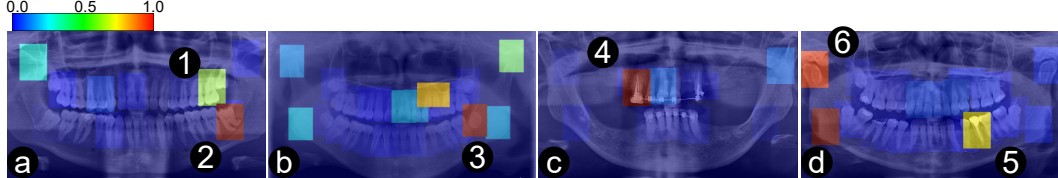

Figure 3: Visualization of the dynamic self-attentions captured by the transformer for four typical instances. Attention rollout is performed to infer the attentions on input patches. View in color and zooming in for the best quality. More attention visualizations can be found in **Appendix**.

meanwhile fixing all other setups. As Table 2 confirms, by consistently achieving the highest accuracy on all metrics with both backbone CNNs, reasoning the relative importance of various attentions is important for representation learning, and the proposed transformer approach can fulfill the goal. More details on PDC, PGFA, Mancs, and the adaptations we performed can be found in **Appendix**.

**Self-attention visualization** We further look into the self-attention reasoning by visualizing raw images and their corresponding self-attention weight maps. In specific, we roll out the attention weights of the transformer encoder layers to capture the propagation of information from input tokens to the final representation [2]. In FoID, each input token is the embedding of a cropped attention patch encoded by the upstream CNN encoder. Figure 3 demonstrates the visualization results of four typical cases. We can obviously see that, in each case, FoID can localize certain identifying anatomies with higher attentions than the others. Such pattern is clear to include tooth with a unique shape (Figure 3(a1)), wise tooth (Figure 3(a2, b3)), tooth with a distinct artifact (Figure 3(c4,d5)), and condyle of uncommon structure (Figure 3(d6)). The phenomenon proves that our method can dynamically compare between the captured attentions, and highlight discriminative ones to boost recognition performance. Moreover, it also shows that the approach has a chance to improve the model explainability, which can be vital when dentists are making final forensic decisions based on deep learning predictions. Nonetheless, we should note that such attention reasoning results is learnt based on our specific training dataset, which can be limited in both size and variety when compared to a real clinical database. As such, more robust attention reasoning can possibly be learnt when the method is applied to large scale databases, and accordingly a higher identification accuracy can also possibly be achieved. We provide visualization results for both success and failure cases of FoID, as well as more detailed analysis, in **Appendix** to help understand the capability of FoID.

**Domain-specific augmentations** We introduce the DSA to generate forensic-realistic views of an instance for the effective instance discriminative learning. To verify the effectiveness of DSA compared to basic augmentations, we perform the ablation study based on both FoID and the CNN baselines. We define a set of basic augmentations (BA) by following existing self-supervised contrastive approaches [44, 10], which include random displacements, rotations, and gaussian noises. More details about BA can be found in **Appendix**. Figure 4 shows that DSA consistently outperforms BA, achieving mAP improvements ranging from 3.28% to 15.32%.

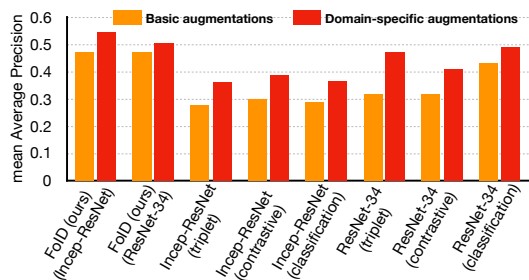

Figure 4: Ablation studies on the proposed domain-specific augmentations for the model training. Various methods are included in comparison. A set of basic augmentations is used as the baseline.

**Objective function design** Three types of losses are included in FoID to complement for the representation learning. In order to understand the effect of each loss type, an ablation study is performed based on both CNN encoders of ResNet-34 and Inception-ResNet. Table 5 indicates that contrastive loss and id classification losses alone can have inferior performance, while the combining of the two leads to largely improved accuracy in terms of mAP (ResNet-34 as backbone: 0.4760/0.3196→0.5394; Inception-ResNet as backbone: 0.4438/0.2808→0.5394). Such phenomenon has been observed by several previous studies on pedestrian re-identification, but the accuracy boosts in our work seem to be more significant

(*e.g.*, 0.817→0.859 and 0.706→0.764 as in [38]). This can possibly due to that two losses in FoID optimize two less correlated tasks of different supports, which makes it more effective for regularizing the transformer. Besides, plugging in the deep supervision signal for the CNN encoder can further increase the accuracy (ResNet-34 as backbone: 0.5394→0.5465; Inception-ResNet as backbone: 0.5394→0.5962), possibly by the improved back-propagation and the direct regularization of the CNN encoder.

## 7  Discussion and Conclusion

Dental forensic identification has always been an important part of criminal/disaster investigation, but is challenging and labour-consuming by relying on expert's visual comparison. Many efforts have been made on systems for storing, accessing and comparing imaging data to benefit the identification, *e.g.*, WinID[3] by The American Board of Forensic Odontology. Our study is the first deep learning approach, aiming to speed up identification in scale and increase the success rate of identifying missing/unidentified/wanted persons, and its inspiring result can possibly lead it to work with the above existing systems.

**Ethical Considerations**   Apart from the promising results achieved by our deep learning solution for dental forensics, its risk of misuse should be considered when deploying such technique. First, although the technique is designed to benefit criminal and disaster investigation, there is a chance for this technique to be used for surveillance. The forensic identification shares a similar nature with pedestrian re-identification — the techniques of which have drawn concerns to intentionally or unintentionally discriminate against marginalized groups. Meanwhile, the pedestrian one can cause identification leakage by one's daily activities captured from surveillance systems, and the forensic one can be by one's medical care experience. As such, medical practitioners should follow the existing medical data protection protocol to avoid the possible identification leakage. Second, forensic practitioners can over-relay on the deep learning results for identification. The proposed method can be limited by the quality of training data as well as the lack of using other medical records entries beyond imaging. As such, we clarify that the method is designed to only provide assistance for dental forensics, rather than make conclusions. An appropriate application of our method can be using the method as an initial recommendation engine for top ranked matches by searching from large-scale medical records among multi-site databases, followed by practitioners manually examining the recommended results for the final conclusions.

**Known Limitations**   There exist two known gaps between our proposed method and the clinical practice of forensic dentists. First, this work only includes panoramic radiographs, since they embody more comprehensive information compared to partial screening, *e.g.*, bitewing x-rays, and is more widely used than expensive 3D screening, *e.g.*, Magnetic resonance imaging. However, in real practice, forensic dentists can take advantages of ***multi-modality*** scans, *e.g.,* by matching certain structures between bitewing and panoramic radiographs. Second, dentists can also utilize ***medical records as priors*** for imaging matching. For example, if a medical record indicates an entity was receiving dental fillings, a shape difference at certain location can thus be expected between the preoperative scan in the database and a query scan. As such, how to design deep learning methods to overcome the two gaps is open to explore.

**Broad impact**   First, this work introduces a novel re-identification task to the machine learning community. Compared to the existing tasks, *e.g.*, pedestrian re-identification, the forensic one differs in several aspects as listed in our formative study. As such, directly applying existing re-identification methods led to inferior results (-9.80% in mAP compared to FoID). Moreover, the needs for the multi-modality awareness and medical priors also make it unique. Combining the potential social impacts, future research might find forensics an interesting research topic. Second, we propose a novel transformer-based self-attention module that enables better attentions aggregation, according to both quantitative and qualitative merits. Such method is also promising for the pedestrian re-identification task, where different attentions (body key-points) can be assigned with different weights for representation based on the relative importance of body parts under certain circumstances (*e.g.,* occlusions). Third, our domain-specific augmentation and multi-task learning have considered domain knowledge of dental forensics. This would be applicable to radiological forensics in general.

---

[3] https://abfo.org/winid/

## 8 Acknowledgments and Disclosure of Funding

We would like to thank the anonymous reviewers for the valuable discussions.

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
