# Appendix for Exploring Forensic Dental Identification with Deep Learning

**Yuan Liang**[1,2], **Weikun Han**[1], **Liang Qiu**[1], **Chen Wu**[1], **Yiting Shao**[1], **Kun Wang**[1]*, **Lei He**[1]*
[1]University of California, Los Angeles
[2]Topaz Labs
liangyuandg@ucla.edu

## 1 Domain-Specific Augmentations (DSA)

We apply the domain-specific augmentations of (*i*) random tooth reduction, (*ii*) random artifact addition, (*iii*) random rigid patch transform, as well as (*iv*) random contrast shifting and Gaussian noise for instance discriminative learning. The DSA is enabled by the anatomical awareness, and its parameters are set by working with dentists to best follow the possible clinical cases. In specific, for random tooth reduction, one tooth area is set to the background intensity according to the tooth mask from semantic segmentation. The background intensity is determined by the average intensity of the non-teeth area. For random artifact addition, two types of artifacts are included: braces and dental filling. In terms of braces, they are modeled as squares mounted at the top [60%, 80%] of the height of a tooth, the length of whose side is set as the 60% of the tooth's width; In terms of dental filling, an area that takes up an area of [30%, 80%] of a tooth height is filled. For both brace and dental filling, a foreground intensity within [220, 256] is randomly selected. For random rigid patch transform, a rotation angle of [-10, 10] degree and displacements of [-15, 15] pixels along x and y axis are set, which represent the common range of transform for orthodontics. For random contrast shifting, the instance intensity is scaled with a random factor from [0.6, 1.4] relative to the instance's mean intensity, and it is clipped to preserve the original intensity range. For random Gaussian noise, the noise mean and std are set to 0 and 0.1 of the maximum intensity of a patch.

All the augmentations are stochastically performed on the fly, and independently for each input attention patch or a whole scan (as in baseline CNN methods). The four types of augmentation are set to take place with a probability of 20%, 20%, 100%, and 100% respectively. Figure 1 demonstrates several randomly selected attention patches before and after processed with the DSA.

**Basic augmentations (BA) for comparison.** We have reported ablation studies on DSA by replacing it with a set of basic augmentation for comparison. We refer to existing instance discriminative learning works of [9, 2] for the implementation of BA. In specific, similar to DSA, random shifts within a pixel range of [-15, 15] are performed both horizontally and vertically; Random rotation angle is chosen within a range of [-10, 10]; Random intensity noises from the Gaussian distribution with a mean of 0 and std of 0.1 of the instance's maximum intensity are added. Different from [9, 2], no image flipping or cropping is applied, since such deformations are rarely seen in the domain of panoramic radiographs. Moreover, no random scaling is performed since different instances of a same entity should maintain a similar anatomies scale.

## 2 Key-Points Localization & Alignment

Two types of key-points are localized in FoID: (*i*) teeth and (*ii*) landmarks of condyle, angle of mandible, and maxillary sinus to avoid over-fit to unrelated regions. To localize teeth, a mask-RCNN [4] for the semantic segmentation of teeth is developed by training and quantitatively evaluating

---

*Corresponding authors

35th Conference on Neural Information Processing Systems (NeurIPS 2021), Sydney, Australia.

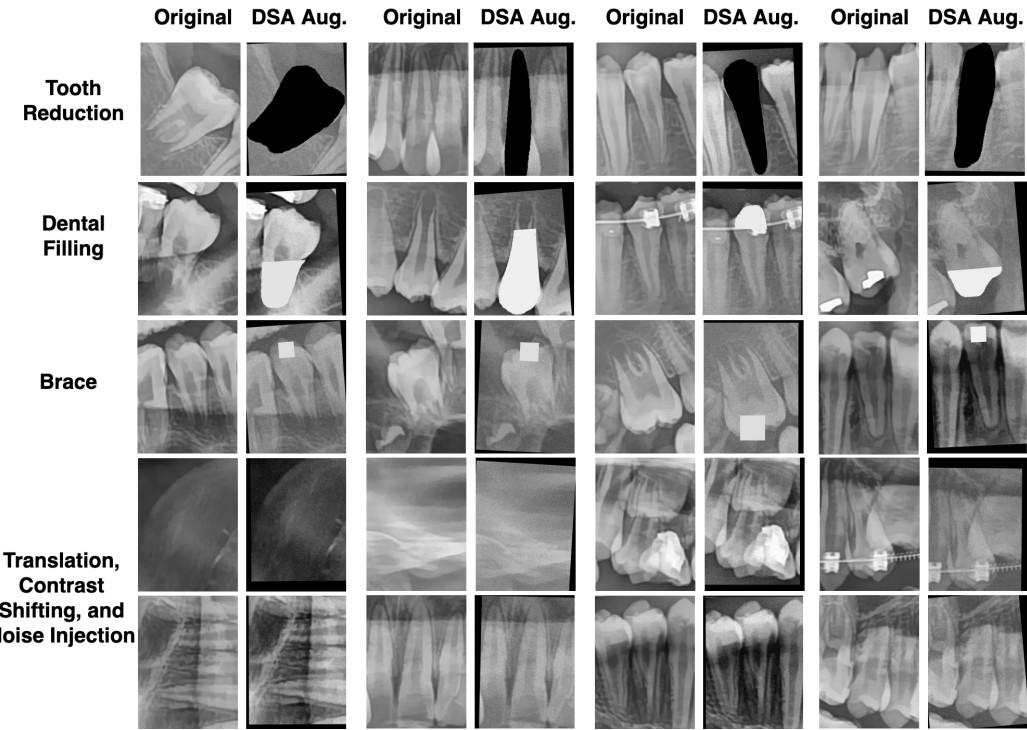

Figure 1: Visualization of the operators of domain-specific augmentations (DSA). The cases are based on randomly selected instances for each type of augmentations.

on the UFBA_UESC dataset by following [7]. The model achieves a mask-wise mean Average Precision (mAP) of 0.705 and a bounding box wise mAP of 0.741, which is comparable to the reported state-of-the-art results [7]. Figure 2 demonstrates the model's predictions with randomly selected cases when the model is applied to our dental forensic benchmark. We can observe that the contours of teeth can be segmented accurately for most cases, meanwhile errors can happen for the classification of teeth as indicated with red arrows in Figure 2. This can be explained as adjacent teeth can have very similar shape and locations. Such errors can lead to the misalignment of attentions. Accordingly, our proposed attention aggregation design of transformer includes no positional information in the input attentions, and assumes no correlations between the importance of attentions and their positions.

To localize the mentioned landmarks, we perform Syn registration [1] to align the atlas to an input instance, and deform the atlas annotation accordingly as the landmark prediction for the input instance. Such method leverages the structure priors of the landmarks, and thus is labeling-efficient: only one pair of scan and its annotation is applied in our study for inference. Figure 3 demonstrates the annotated case, as well as the key-point localization results of randomly selected cases. For each case, the deformed atlas, deformation map, and the predicted landmarks over the target scan are illustrated. We can clearly see that the approach achieves satisfying results for most cases, which validates its usage for the purpose of attention cropping. Meanwhile, red arrows indicate the places where large localization divergences happen. Such errors can be caused by both: (*i*) large variations of the underlying dental structures, and (*ii*) large variations in the vision field of the mouth cavity due to the different scanning setups. To reduce such errors, multi-atlas localization approach can be applied in the future [6]: it selects multiple representative atlases from different populations and scanning setups, and fuses the localization results from those atlases for the final prediction.

## 3   Identification Results & Self-Attention Visualization

Figure 4 demonstrates the FoID's retrieval results of both success (*i.e.,* matching instances from the gallery are ranked 1st in the retrievals) and failure cases (*i.e.,* matching cases from gallery get lower

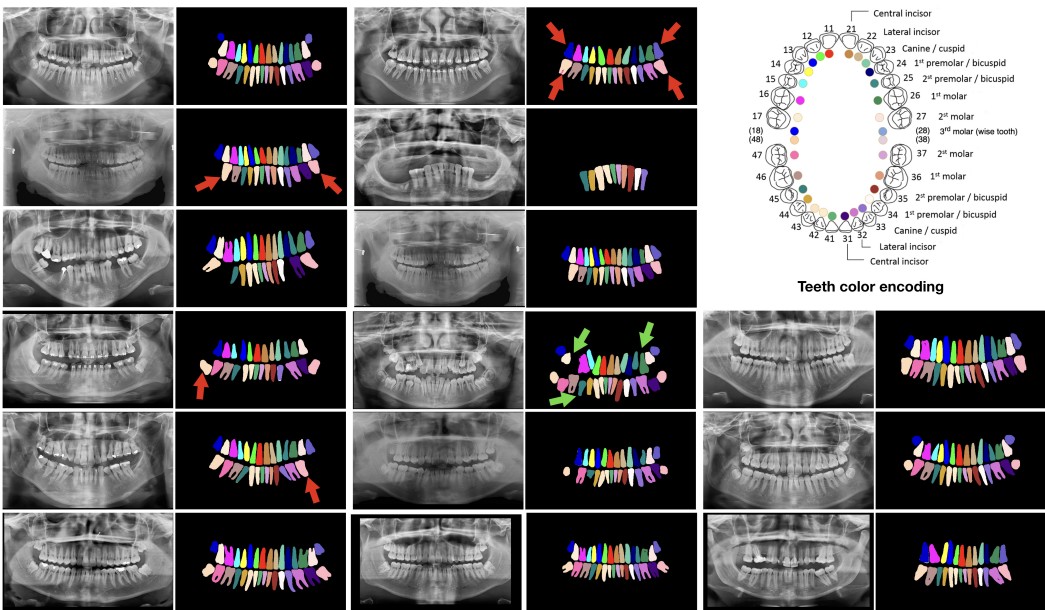

Figure 2: Results of the semantic teeth segmentation for randomly selected cases from our benchmark. Red arrows indicate the wrong teeth classification, while green arrows indicate the missing teeth in segmentations. Top left shows the color encoding protocol for each type of teeth.

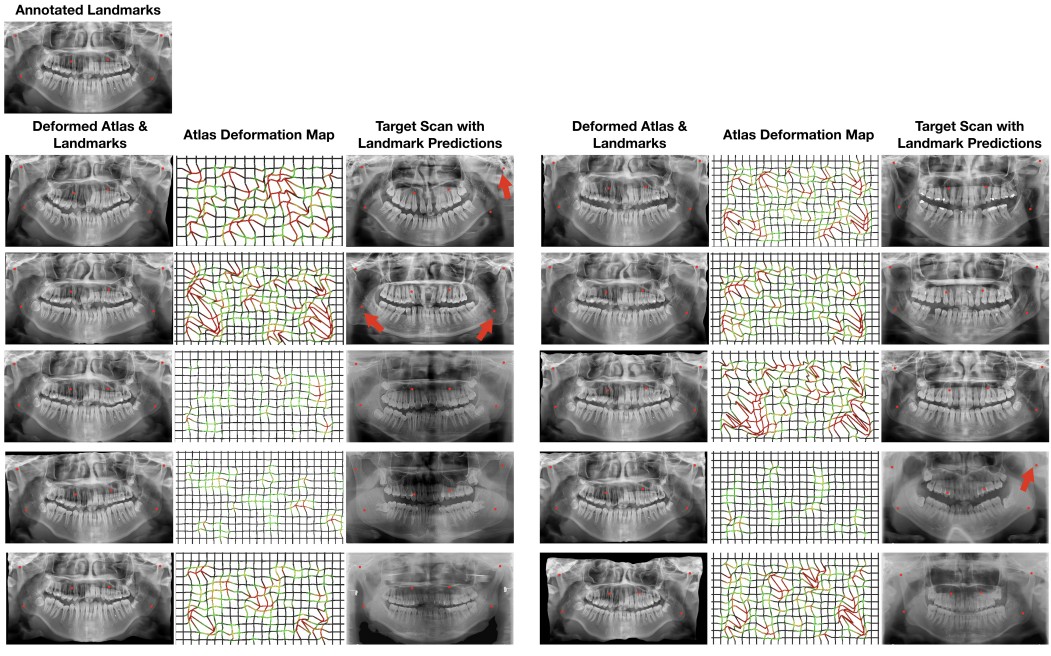

Figure 3: Results of the registration-based landmark localization. Top left shows the annotated atlas scan with the six landmarks of condyle, angle of mandible, and maxillary sinus. The following rows show the randomly selected localization results from our method. For each case, (*i*) the deformed atlas, (*ii*) atlas deformation map, and (*iii*) the landmark predictions over the target scan are illustrated. The landmarks are shown with red dots. The results clearly show that the landmark predictions are mostly accurate for the purpose of attention cropping. Red arrows show the failure cases where landmark predictions are largely deviated from the accurate locations. View in color and with zoom-in for the best quality.

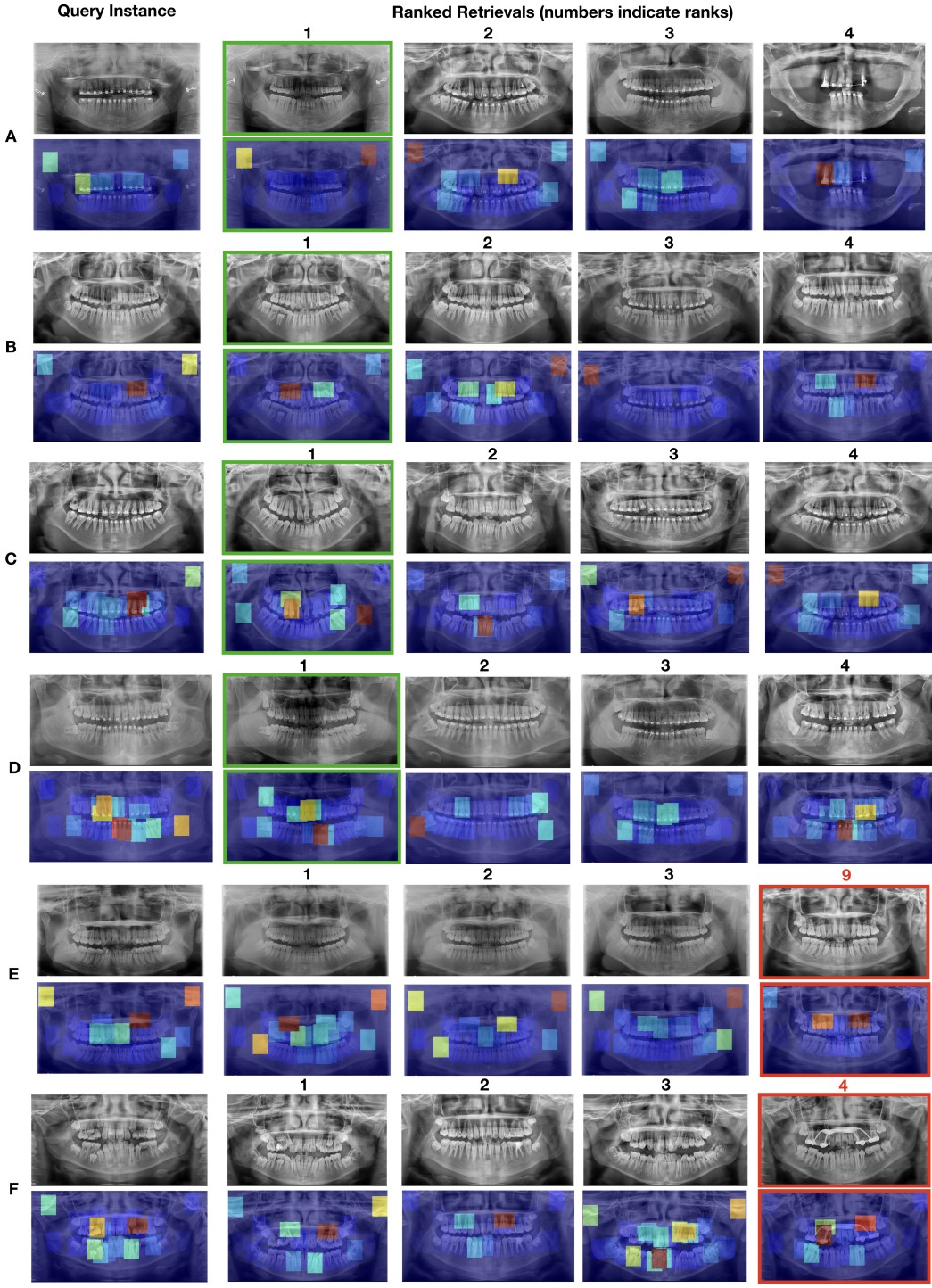

Figure 4: Randomly selected success and failure cases of FoID's retrieval. The first column shows the query instance; the following columns show the retrieved cases from the gallery as well as their ranking in the retrieval (the digit above each image). Rows A-D are success cases where the matching instances are ranked 1st within the retrievals (marked with green borders). Rows E-F are failure cases where the matching instances are ranked lower (marked with red borders).

ranks within the retrievals). Moreover, the captured attention maps from the transformer are also visualized in order to provide insights into the retrieval results. Note that all the query instances are randomly selected from the testing set of the benchmark. The results are generated from the best model of FoID, which utilizes the Inception-ResNet as the CNN encoder is applied, achieving the mAP of 59.62% and rank-1 accuracy of 50.56%.

Figure 4(A,B,C,D) demonstrate the success cases, where we can clearly see the following patterns. (*i*) Our proposed self-attention module can capture the relative importance of anatomies dynamically based on the input instance. That is, different instances can have distinct attentions on the anatomies. (*ii*) The anatomies that are out of the general appearance distribution among populations can gain larger attention from the transformer. Such anatomies can include those with artifacts (*e.g.,* (A, rank-4), (D, rank-1), (F, rank-4)), teeth with unique shapes (*e.g.,* (C, rank-1), (C, rank-4), (D, rank-1)), and landmarks with unique structures (*e.g.,* (A, rank-1), (B, rank-1), (E, rank-2)). (*iii*) The correct alignment of attentions between the query instance and the matching instance can contribute to the high accuracy of FoID. By looking into the query and rank-1 instances in cases A-D, we can observe such consistency in the attentions. Moreover, the case A and D show that, although there can be large differences in the appearances of same anatomies between instances (*i.e.,* variations caused by braces as in the two cases), FoID can still get the correct retrieval. This can be explained as the CNN encoder is applied and effectively trained for extracting robust features for the anatomies. For the cases E and F where the matching instances are not correctly retrieved, we can clearly see that there exists certain similarities between the query instances and the wrongly retrieved instances. Such error can be caused by the lack of captured discriminating anatomies and the mismatching of the attentions.

## 4 Ablation Studies on Attention Mechanism

We present the implementation details of the three state-of-the-art attention mechanisms that FoID is compared to: PDC [10], PGFA [8], and Mancs [12]. Since the methods are originally developed for pedestrian re-identification, we make the following necessary modifications to the models and training strategies such that they can work with our task.

- To reproduce PGFA [8], we replaced the 18 channel heatmap with the 38 channel key points segmentation map. While the original heatmap utilizes a soft attention mechanism, we applied the hard attention mechanism here since the uncertainty in the anatomies is much less in our case. We referred to the source code from `https://github.com/lightas/ICCV19_Pose_Guided_Occluded_Person_ReID`.

- For the implementation of Mancs [12], we closely followed the source code from `https://github.com/hustvl/mancs`, which is under the MIT License.

- For PDC [10], we switched the backbone model from GoogLeNet [11] to ResNet-50 for keeping consistent with the PGFA and Mancs. An input image with a size of H×W will result in an output feature maps with a spatial dimension of H/16×W/16. Moreover, we discarded the Feature Weighting sub-Net in PDC [10] to preset no assumptions on the relative importance of the 38 key points. We referred to the code from `https://github.com/ljn114514/PDC-ICCV2017`.

Overall, the version of ResNet-50 [5] pre-trained on ImageNet [3] is used as the backbone for all the three methods. The final embedding has a size of 2048, and is extracted from the last average pooling layer in the backbone. The input image is resized to 512×512 with padding. The batch size is set to 6 during training, and the repeat of online augmentation is set to 4 per sample for all the methods. All the models are optimized using RMSprop with a constant learning rate of 1e-6, a weight decay of 1e-4, and a momentum of 0.9. All experiments are based on three GeForce GTX 1080 Ti GPUs.