# OpenReview forum: "Exploring Forensic Dental Identification with Deep Learning"
_NeurIPS.cc/2021/Conference — NeurIPS 2021 Poster_

### Official Review · Reviewer_tMZm · 2021-07-14

**Rating:** 7
**Confidence:** 3

**Summary:**

The work proposes to use deep learning for forensic dental identification, i.e., identifying living or deceased persons from their panoramic dental radiographs. It outlines the main, domain-specific challenges: lack of paired radiographs for metric learning, lack of annotations (e.g., bounding boxes of salient anatomic parts used by dentists), and visual variability. The paper proposes the new task, describes a set of domain-specific augmentations (based on anatomical variation), and suggests strategies to adapt existing networks to this domain.

By introducing forensic dental identification as a new ML task, the authors point out a new problem that could invite other researchers to design algorithms for.

**Ethical Concerns:**

EDIT (08/18): see main review.

**Limitations And Societal Impact:**

While the paper proposed a very novel ML application, it is also very limited to this application. This is an inherent limitation and while there is little the authors can do about this, they should discuss more specifically how their findings can affect other tasks such as person re-identification, data augmentation etc.

I appreciate the thorough description of the limitations and broader impact. However, due to privacy concerns, how likely would the proposed system be used in practice?


**Main Review:**

The paper proposed an interesting and novel task, and designs a state of the art network to solve it. I found the introduction and related work easy to understand and very insightful. The results section, however, was difficult to follow and had quite a bit of grammar issues. Given that the architectural findings that the authors discovered, what implication does this approach have to forensic dental identification and machine learning in general?

Here are some specific questions:

- How important is it to use a panorama-retrained model?
- Please add visualizations and sample annotations of regions used for hard attention
- Please include a visualization of the proposed domain-specific augmentations
- Include an analysis of false negatives, as this may be an important limitation of using this approach. How is the performance in comparison to identification performed by humans?

I was not able to download the datasets, and do the authors plan to release it? I think this would be a critical task because the goal of the paper is to encourage more exploration in this specific domain.

Typos/corrections:
- Line 15: “effeteness” -> effectiveness
- Line 47: “In specific,…” -> bad grammar
- Line 147: “In specific” -> Specifically
- Line 152: “does not relay” -> does not rely
- Line 212: “we incorporates” -> we incorporate
- Line 270: “few study” -> few studies
- Line 312: “We can obviously see” -> We can see that. Why obvious?
- Line 325: “effeteness” -> effectiveness


EDIT (08/18): I have read the other reviews and authors' rebuttal. I thank the authors for clarifying my concerns. I second other reviewers' concern about adding additional discussion about ethics, limitations, and data description. Please see main comment. I do, however, think that this paper will be very valuable to the ML community as a practical application of current technologies to a new task.

**Time Spent Reviewing:**

4

---

> ### Author Response · Authors · 2021-08-10
> **Response to Reviewer tMZm**
>
> Thank you so much for your valuable comments!
>
> Q1: What implication does this approach have to forensic dental identification and machine learning in general? Discuss more specifically how the findings can affect other tasks?
>
> A1: We explore and verify: 1. the domain-specific augmentations for learning from unpaired data, 2. prior knowledge-guided attention for learning from information-sparse data, and 3. self-attention mechanisms for reasoning significant regions for prediction. Our study is the first to show that these techniques have good potential for the broad but underexplored area of radiological forensics. Hopefully, our study can draw attention from the machine learning community by introducing deep learning to a new task forensic identification, obtaining nice initial outcomes, as well as illustrating novel challenges and large room for improvement.
>
> Our study is different from other identification tasks (e.g., pedestrian re-identification) in several aspects as listed in our formative study --- directly applying existing identification methods led to inferior results in our experiments. Considering the potential social impacts, machine learning for forensic can potentially be a new niche for research and forensic practice in the future.
>
> We will add more related discussions in the revised paper.
>
>
> Q2: How important is it to use a panorama-retrained model?
>
> A2: The panoramic image is preferred for two reasons according to the practitioners. First, compared to partial screening, e.g., bitewing x-rays, panorama x-ray embodies comprehensive information about the oral cavity, thus benefits identification. Second, compared to 3D screening, e.g., MRI, panorama x-ray is cheaper, and thus more widely available in medical records for matching. Our future work will utilize other modalities as complementary to improve accuracy.
>
>
> Q3: Adding visualizations and sample annotations of regions used for hard attention
>
> A3: The visualizations for the annotations as well as predictions can be found in the Appendix section 2.
>
>
> Q4: Including a visualization of the proposed domain-specific augmentations
>
> A4: The visualizations for the domain-specific augmentations can be found in the Appendix section 1.
>
>
> Q5: Including an analysis of false negatives to reveal the limitation. How is the performance in comparison to humans?
>
> A5: The visualizations for false negative cases and their analysis on self-attention can be found in Appendix section 3 (Identification Results and Self-Attention Visualization). We did not compare the accuracy to manual study by practitioners, and this is the immediate next step for our future study.
>
> Q6:Plan to release the dataset?
>
> A6: To encourage future research on the forensic topics, we will release a portion of the test set upon paper publication.
>
>
> Q7: Due to privacy concerns, how likely would the proposed system be used in practice?
>
> A7: The forensic dental identification has been long pursued. Examples include WinID (https://abfo.org/winid/) by The American Board of Forensic Odontology and NICI (https://www.fbi.gov/services/information-management/foipa/privacy-impact-assessments/national-dental-image) by FBI. These are existing systems for storing, accessing and comparing imaging data for law enforcement. However, no widely accepted automated solution has been developed, and the above existing systems rely on the manual comparison of records. Technically, our proposed method can be combined with the above existing systems without introducing any new concerns on privacy. This can be achieved by collaborations with the related government agencies and non-government organizations.

---

### Official Review · Reviewer_zijN · 2021-07-15

**Rating:** 5
**Confidence:** 4

**Summary:**

The authors propose a framework for retrieval of dental panoramic X-Rays.
To find regions of interest they focus on teeth which are extracted by semantic segmentation using a pretrained mask-RCNN as well as relevant areas extracted based on landmarks for which it is necessary to register every image to an atlas in a deformable way. All those ROIs are extracted and augmented in a domain-specific way (fillings, tooth extraction, etc), which serves as input to a CNN which is trained with an objective function that consists of three loss terms, added in an unweighted manner: a contrastive loss, an ID loss and a deep supervision loss.

They use a publicly available dataset for training, but test the proposed method on data collected by the authors, s.t. each image in the test set is to be retrieved from the union of the test and training set.

An ablation study on different modules of the framework is performed to test their contributions to the overall improvement of the proposed method.


**Limitations And Societal Impact:**

Yes

**Main Review:**

The framework combines multiple state of the art methods in order to solve a specific and very challenging task.
It is interesting work in that there are several challenging subtasks that needed adaptation from their original publications to this kind of dental data which all come with their own challenges and hyper parameters (starting from the deformable registration to an atlas and the semantic segmentation to generate the input data).
However I am missing a novelty which is transferrable to other kind of data than this explicit application. I believe that this work is very valuable and of significant importance in the field of digital dental identification, but I am not convinced that it entails enough novelty from an ML perspective that is of general purpose to other fields of application. (If the authors can point out a novelty which is of a broader interest apart from dental identification, then this will also have to be reflected in the broad societal impact.) As the framework consists of many subtasks, there is only little space to provide all details needed for reproducibility in a conference paper like this and I believe a journal which is in theme closer to the paper’s application would be a more suitable publishing channel.

Questions to address:


It is unclear to me how the baseline experiments were conducted which are presented in Table 2. What exactly was done for e.g. ResNet-34+reconstruction, or ResNet-34+triplet, etc.? What was the pipeline for one such experiment?

I have some questions regarding the dataset. The paper says that UFBA_UESC was used to pretrain the mask-RCNN (line 132) for the semantic segmentation step to locate the teeth. Later, for the benchmark construction, the DNS Panoramic dataset is used as a training set. In [50], in which the DNS Panoramic dataset is introduced, it says that the original dataset, named “UFBA-UESC Dental Images data set” (published by Silva et al., ‘“Automatic segmenting teeth in x-ray images: Trends, a novel data set, benchmarking and future perspectives”, 2018) was modified by Jader et al. (“Deep instance segmentation of teeth in panoramic x-ray images”, 2018) to allow for instance segmentation, resulting in the dataset named “UFBA-UESC Dental Images Deep”. Silva et al. in [50] (“A study on tooth segmentation and numbering using end-to-end deep neural network”, 2020) then modified “UFBA-UESC Dental Images Deep” further, resulting in the “DNS (Detection, Numbering, and Segmentation) Panoramic Images”, which in turn was used as a training set in the proposed publication. So I wonder in what ways the UFBA_UESC dataset used for pretraining in the mask-RCNN step differs from the DNS dataset used for the overall training, and if the pretraining - in case of a big overlap of the two datasets - has much/any impact?

What size are the images in pixels and how do they need to be resized to fit the CNN encoder which was pretrained on RGB ImageNet images?
Is the atlas publicly available to which the images get registered?

Regarding the addition of the three types of loss in the final objective function: are they all of the same magnitude, or is one of the loss terms dominating over the other two?

Is the test set released?

What is meant by online mining on line 201 in this context?

Minor comments: formatting line 301;  typos or similar in lines: 143, 169, 177, 181, 184, 206, capitalization of embedding dim. line 207, 212, 224, 270/271, 282, 405,


**Time Spent Reviewing:**

10

---

> ### Author Response · Authors · 2021-08-10
> **Response to Reviewer zijN**
>
> We really appreciate your valuable comments!
>
> Q1: Novelty of a broader interest apart from dental identification? Novelty that is transferable to other kinds of data than this explicit application?
>
> A1:  See A1 to Q1 in the response to Reviewer #2 (kU4j).
>
>
> Q2: How the baseline experiments, e.g., ResNet-34+contrastive, were conducted in Table 2 in section 6.2.
>
> A2: The baseline models follow [A metric learning reality check, Musgrave, 2020] by using standard training objectives without attention mechanisms. Specifically, ResNet-34+contrastive takes in a resized and padded image, encodes it with ResNet34, and optimizes the whole model with contrastive loss. Besides, our domain-specific augmentations are also applied to enable training with unpaired data.
>
> The experiments are different from those in section 6.3, where several attention mechanisms from pedestrian re-identification works were applied for comparison.
>
> We will make it clearer in the revised paper.
>
>
> Q3: The overlap in datasets (UFBA_UESC and DNS) for training teeth segmentation mask-rcnn and the identification model.
>
> A3: This has little impact on the results as explained below.
>
> Mask-rcnn is trained for teeth segmentation, the results of which are used as one of the inputs to our identification model. Denote the set for learning teeth segmentation as A, and the set for identification as B = {B1, B2}, where B1 is unpaired data for training, and B2 is paired data that are unseen by the training, but have matches with the query. There are 271 images that overlap between A and B1, and no overlap between A and B2. There should be little impact here since:
>
> 1. Segmentation model is unseen by the testing data of B2. This would not artificially increase accuracy during the test.
>
> 2. This setup is similar to the real forensic application according to dentists: forensic practitioners will possibly label a subset of available data (in the repository) for training the teeth segmentation, resulting in a same overlap.
>
> Technically, the mask-rcnn can be trained with any teeth segmentation labeling datasets based on the data availability.
> We will clarify this in the revised paper.
>
>
> Q4: Image sizes in pixels and how do they fit into the CNN encoder?
>
> A4: All images are unified by the same pixel spacing (0.11mm per pixel). Since they are from different machines, they can have different dimensions in pixels, but the mean numbers are around 2500 x 1700. Each image is cropped for patches of a fixed size of 256 x 196, guided by attention, and stacked to feed into the CNN encoder.
>
>
> Q5: Is the atlas publicly available to which the images get registered?
>
> A5: Not publicly available right now. but we will release the atlas pair along with annotations upon publication.
>
>
> Q6: Are three types of loss in the final objective function of the same magnitude?
>
> A6: According to our observation, yes. We will provide a visualization of the training curves in the appendix in the revised paper.
>
>
> Q7: Is the test set released?
>
> A7: To encourage future research on the forensic topics, we will release a portion of the test set upon our paper publication.
>
> Q8: What is meant by online mining in section 4.4 objective function and training?
>
> A8: Mining here aims at finding the best image pairs to train the model. Here the online mining means the pairs are selected within a randomly sampled batch, rather than considering the pair selection during the batch sampling. This is known to be the sota practice as discussed in [A metric learning reality check, Musgrave, 2020].

---

> > ### Comment · Reviewer_zijN · 2021-08-17
> > **Addressing the Authors' response**
> >
> > I thank the authors for the addressing my questions!

---

### Official Review · Reviewer_kU4j · 2021-07-16

**Rating:** 4
**Confidence:** 4

**Summary:**

The authors propose a new architecture FoID for forensic identification of individuals from dental imagery, a task often conducted by domain experts for identification at crime scenes/disaster sites. Given an image and a tooth segmentation map, they augment a straightforward CNN in four fashions: (i) hard attention on the regions of the scan where identifying information can be found, (ii) dental-specific augmentations (e.g. addition of a cavity) as opposed to general augmentations, (iii) the inclusion of three loss families (classification, contrastive, identification), and (iv) a transformer-based attention mechanism for focusing on unusual shapes of interest. In experiments, they outperform a series of benchmarks,  and ablations show the importance of each of the augmentations above. They include interpretations of the self-attention mechanism for explainability.

**Ethical Concerns:**

As mentioned above, there are some ethical concerns with re-identification (particularly in non-dental settings they propose FoID could be extended to). While I don't think it necessarily requires an ethical review (though I'm not sure the line), I do think it requires at minimum further discussion from the authors.

**Limitations And Societal Impact:**

I do think a little more emphasis should be placed on discussing the privacy risks of re-identification, particularly in settings outside of dentistry. While dental images may be locked within medical records, the abstract says the proposed techniques could be repurposed for pedestrian re-identification, and given there is currently a large debate on the ethics around person re-identification, particularly due to bias, I think it is important authors give the issue more consideration in the paper.

**Main Review:**

The authors present an architecture that seems quite effective for forensic identification, on a relatively novel task for the machine learning community. No proof/theoretical claims are provided, and all empirical claims seem well-founded with fair and thorough baseline comparisons across multiple CNN architectures and ablations. The domain-specific augmentations are clever and thorough in creating effective data pairs. The provided attention maps further seem to indicate that the model is operating as expected. The paper is clearly written and easy to follow, and I appreciate the author's inclusion of code in their submission.

However, there is admittedly limited novelty of the methods and conclusions from this work , which may not make it most suitable for the NeurIPS community. Going through the contributions:
* Image augmentations have long been used as a way to augment data sets, and it seems expected that the more hand-crafted, tailored augmentations for the dental domain would outperform out-of-the-box standard approaches.
* It is relatively standard to have multiple objective/objectives types in the machine learning community, and the authors themselves include the fact that the advantage of multiple objectives has already been shown in the closely related pedestrian re-identification domain.
* While the transformer self-attention mechanism is in a slightly different position in the network than might be standard, there is now a wealth of work using transformer self-attention mechanisms (e.g. https://arxiv.org/pdf/2101.01169.pdf), including using them in combination with CNNs.
As a result, it's not clear to me whether the paper necessarily provides generalizable conclusions beyond what's already existing in recent literature.

As a minor note to the author, you likely mean to use “effectiveness” over “effeteness”

**Time Spent Reviewing:**

4

---

> ### Author Response · Authors · 2021-08-10
> **Response to Reviewer kU4j**
>
> We really appreciate your valuable comments!
>
> Q1: Limited novelty of the methods/conclusions making the work not suitable for the NeurIPS community.
>
> A1: Our work is the first to investigate deep learning in detail to a new field, forensics. Application of machine learning to new fields for social impacts, instead of methodology innovation, has been the primary contribution for many NeurIPS papers. Examples can include [SEVIR : A Storm Event Imagery Dataset for Deep Learning Applications in Radar and Satellite Meteorology, Veillette, 2020], [Audeo: Audio Generation for a Silent Performance Video, Su, 2020], and [MRI Banding Removal via Adversarial Training, Defazio, 2020]. The presented task in our work is unique compared to existing re-identification tasks (e.g., pedestrian re-identification) in several aspects as listed in our formative study.  Directly applying existing re-identification methods led to inferior results, but our study has led to much better results (+9.80% in mAP). Moreover, the needs for the multi-modality awareness and medical priors as explained in the discussion section are also novel. Combining the potential social impacts, future machine learning research might find forensic an interesting research topic.
>
> Regarding the novelties in methodology, we propose a self-attention mechanism for better feature aggregation by leveraging the relative importance of patches, according to both quantitative and qualitative merits. We plan to verify and extend the approach for other re-identification tasks and for multi-instance learning in the future. For audiences from medical research, our domain-specific augmentation and multi-task learning have considered domain knowledge of dental forensics. This would be applicable to radiological forensics in general, which is a mostly unexplored area for machine learning for now.
>
>
> Q2: Privacy risks and ethical issues of the study, particularly in settings outside of dentistry.
>
> A2: The most significant contribution of our work is about dental forensics, and radiological forensics in general. Regarding ethics, distinct from the existing person re-identification with nature photos relying on faces and clothing, the radiological re-identification relies on the imaging of the interior body, which can lead to reduced bias. Regarding privacy, the research of forensic identification should strictly follow the anonymous processing steps for data, while the application of the algorithms should be subject to the existing protection protocols for medical data. When the proposed methods are applied outside radiological data, bias and privacy should take more attention: e.g., when introducing prior knowledge with the domain-specific augmentations, the inclusion of knowledge should be carefully designed to avoid bias.
>
> We will add more detailed discussions in the revised paper.

---

> > ### Comment · Reviewer_kU4j · 2021-08-26
> > **Response**
> >
> > I realize I didn't respond here, but I just wanted to let you know that I read and considered your rebuttal and have entered in conversation with other reviewers. Thanks for the response both to me, and the update to the ethic reviewer; I have no further clarifications at this time!

---

### Official Review · Reviewer_t8rG · 2021-07-16

**Rating:** 7
**Confidence:** 4

**Summary:**

The paper presents an extensive study on applying deep learning (including novel techniques) to dental forensic identification.

**Main Contributions:**

- An in-depth discussion on the challenges of the task and limitations of past works;
- Multiple techniques ranging from data augmentation to network architecture to tackle these challenges (with potential applications to other tasks);
- A comprehensive dataset covering many dental variations, to reflect a realistic scenario;
- An extensive and thoughtful ablation study that gives insights on the effectiveness and benefits of each major technical decision or proposed technique; and
- SOTA results in the task by a large margin vs. sensible baselines.

**Ethical Concerns:**

Since some data sources are anonymized (maybe for the review process? see L227), we can't evaluate if there are any ethical concerns there. Considering this is medical data, this might be problematic. The paper does claim the data is anonymized.

**Ethics Review Area:**

["I don’t know"]

**Limitations And Societal Impact:**

The paper acknowledges and addresses the limitations of the proposed dataset, as well as the privacy risks of dental identification (which, like they discuss, are less risky vs. person reid and facial recognition). It might have been helpful to further discuss the potential impact of this technique in the context of law enforcement, which is prominent in this paper.

**Main Review:**

The paper presents extensive and thoughtful study on deep learning applications to dental forensic identification.

First, it presents a discussion on limitations of past work and challenges of this task, encompassing both the available datasets and previously used models. This discussion is grounded on insights from dental professionals, addresses short-comings from techniques from related fields (e.g. person reid), and lays the foundation from the proposed techniques. Additionally, the ablation studies carefully confirm these claims, with the proposed solutions individually outperforming across the board.

Second, it demonstrates a novel dataset that addresses many of the limitations of past work, e.g. by including many dental variations and scenarios. Certain scenarios that are not covered in the dataset are discussed later in the paper.

Third, it proposes a novel approach to leveraging deep learning to tackle this task. The techniques proposed range from data augmentation to network components. It cleverly combines approaches from diverse existing literature with original contributions, with justifications based on insights from dental professionals, and accompanying comprehensive ablation studies to validate their hypotheses.

Finally, it presents a careful evaluation of the work, including comprehensive ablation studies. The proposed technique achieves SOTA results that strongly outperform sensible baselines presented by the authors. The ablation studies corroborate the choice of the major individual components of the technique (e.g. data augmentation, objective function).

The writing is generally clear, and the code is available. The dataset is based on a publicly available source, and another anonymized source (I believe for the review process).

All in all, the paper presents a thoughtful discussion, novel methods (with potential applications to other tasks), and a careful and comprehensive evaluation.

---

Minor:

- L301: missing/broken citations ("[]")
- L15: typo "effeteness"



**Needs Ethics Review:**

Yes

**Time Spent Reviewing:**

4

---

> ### Author Response · Authors · 2021-08-10
> **Response to Reviewer t8rG**
>
> Thank you so much for your valuable comments!
>
> Q1: The potential impact of this technique in the context of law enforcement.
>
> A1: The forensic identification with radiology is vital in the enforcement of civil and criminal law by investigating human remains. Existing efforts include WinID (https://abfo.org/winid/) by The American Board of Forensic Odontology and NICI (https://www.fbi.gov/services/information-management/foipa/privacy-impact-assessments/national-dental-image) by FBI. These are systems for storing, accessing and comparing imaging data for law enforcement. However, no accepted automated solution has been developed, and the above systems still rely on manual comparison of records.
>
> Our study is the first deep learning approach, aiming to speed up identification in scale, and increase the success rate of identifying missing, unidentified, and wanted persons. Moreover, the research outcome can possibly increase fairness than manual comparisons during law enforcement. Besides, although our method is experimented with dental x-ray, challenges are most common for other imaging modalities (e.g., MRI) and other commonly used body anatomies (e.g. skeleton, another enduring body structure). As such, our solutions can help forensic research in general.
>
> We will add further discussions in the revised paper.
>
>
> Q2: Missing citations and typos.
>
> A2: We will add the missing citations and correct the typos in the revised paper.

---

> > ### Comment · Reviewer_t8rG · 2021-08-17
> > **Response to rebuttal**
> >
> > Thank you for addressing my questions and incorporating the comments in the revised version.

---

### Review · Ethics_Reviewer_vxyR · 2021-08-12

**Recommendation:**

Many of the concerns are relatively easy to address in additional evaluation experiments and in a more thorough discussion of ethical considerations. But I would recommend to ensure that these concerns have been addressed in the paper before accepting it.

**Ethical Issues:**

Yes

**Ethics Review:**

The paper introduces an approach to forensic dental identification. It uses a (previously published) dataset of dental images, and every person's data is recorded over time. All the data has been anonymized.

There are important use cases for the model (e.g. criminal/disaster investigation) but also high risk of misuse of the method. There are several possible misuse scenarios, e.g. (1) for surveillance (as is evident from the related work overview in the paper, and multiple references to pedestrian re-identification task), and (2) a risk of over-reliance on the method by people who do not understand the method's limitations. Despite outperforming other CNN-based approaches, current best mAP results are 0.59, and it is not clear how these automated approaches compare to human assessments. In people-centric tasks like the one discussed in the paper, the cost of an error can be higher than in less sensitive applications, and this issue needs to be discussed explicitly in the Ethical Considerations section, so that potential (non-technical) users will be aware of the risks.

Another concern is that it is not clear which biases the model perpetuates and amplifies: the datasets are not balanced across demographics (gender, age) and they might have worse results for specific slices of populations.

Since there are concerns for mis-use and for biases, I would recommend to include (1) a more thorough people-centric analysis of results, e.g. presenting results across demographics, to show that the model does not have divergent mAP results across populations, to control for biases; (2) a more thorough discussion of ethical considerations, including the potentials for mis-using the model.

---

> ### Author Response · Authors · 2021-08-19
> **Response to Ethic Reviewer vxyR**
>
> Thank you for the valuable comments from the perspective of ethics! We will extend the discussion accordingly in the revised paper. We hope it can help elite more meaningful discussions about ethics withinin the ML community.
>
> 1. Results across demographics
>
> We will report the accuracy across genders and two age intervals (<40 and >=40) of our models in a new table in Section 6.2. The given extra page will provide enough space for the table on paper’s acceptance.
>
>
> 2. Thorough discussion of ethical considerations
>
> We agree that our proposed deep learning solution can have a risk of misuse. We will add a section named ethical considerations for detailed discussions on the risk. The given extra page will provide enough space for this section on paper’s acceptance. The section will include the following main content:
>
> Apart from the promising results achieved by our deep learning solution for dental forensics, the  risk of misuse should be considered when deploying the technique. First, although the technique is designed to benefit criminal and disaster investigation, there is a chance for this technique to be used for surveillance. The forensic identification shares the same nature with pedestrian re-identification: the techniques of which have drawn concerns to intentionally or unintentionally discriminate against marginalized groups. Meanwhile, the pedestrian one can cause identification leakage by one’s daily activities captured from surveillance systems, and the forensic one is by one’s medical care experience. As such, medical practitioners should follow the existing medical data protection protocol to avoid the possible leakage. Second, forensic practitioners can over-relay on the deep learning results for identification. The proposed method can be limited by the quality of training data as well as the lack of using other medical records entries beyond imaging. As such, we clarify that the method is designed to only provide assistance for dental forensics, rather than make conclusions. An appropriate application of our method can be using the method as an initial recommendation engine for top ranked matches by searching from large-scale medical records among multi-site databases, followed by practitioners manually examining the recommended results for the final conclusions.

---

> > ### Comment · Ethics_Reviewer_vxyR · 2021-08-20
> > **Response to Authors**
> >
> > Thank you for your response. If your data is also labeled along race axis, I would recommend to include this attribute, in addition to gender and age.
> >
> > Overall, your proposed modifications -- reporting ablation studies across demographics, and incorporating a detailed discussion of potential risks in the Ethical Consideration section -- would address my concerns. This is provided that the new results across demographics will not reveal biases against specific populations. If you do reveal significant disparities, I would recommend not to publish the paper until your model works equitably across populations.

---

### Review · Ethics_Reviewer_Ws2D · 2021-08-13

**Recommendation:**

I urge the authors to expand the limitations discussions as well as add more discussion of potential misuses. I also urge the authors to add more details about the data sources (I assume this is due to the anonymyzation requirement)

**Ethical Issues:**

Yes

**Ethics Review:**

The paper presents a benchmark dataset as well as deep learning based method to for dental forensic identification, with applications to criminal/disaster investigations.

Although the ethical review was flagged by a reviewer because of the lack of transparency about the data sources, I find the paper could do a much better job of discussing broader issues associated with this technology, that goes beyond the transparency about sources.

First, since you are proposing a benchmark dataset, especially within a medical domain, it would be great to understand what sociodemographic populations are represented in the dataset. While the number of identities present in the data is relatively small, it still begs the question of representational diversity. This is especially important since you are proposing this as a benchmark to measure progress on this task against.

Secondly, the current limitations and social impact section could be further expanded to include what are the potential misuses or malicious uses of this technology. The authors do list an array of potential downstream use cases, many of them with potential ethical issues. While it is totally ok to allude to potential downstream applications, it would be great to discuss the failure modes more carefully.

---

### Author Response · Authors · 2021-08-10
**To all reviewers**

We thank all reviewers for their very detailed and constructive comments. We are very encouraged by the comments that the work introduces a novel and challenging task for machine learning (R1, R2, R3, R4), with insightful discussions (R1, R4) grounded on knowledge from practitioners (R1); the work achieves SotA by a large margin compared to strong baselines with several proposed techniques (R1, R2, R3, R4), and carefully verifies the techniques with sufficient experiments (R1, R2, R3). We hope our work would inspire machine learning research into forensics, an important but underexplored area, and the proposed method could be extended to other identification tasks.

---

### Comment · Reviewer_tMZm · 2021-08-18
**Post-rebuttal comments**

I have read the rebuttal and other reviews. I agree that there are some ethical concerns with regard to not releasing the test dataset and associated patient demographics, as well as many concerns with potential misuse of the proposed technology for surveillance and similar tasks. As a result, authors promised to release a portion of the test set and detail the issues that were raised in the paper.

In any case, these ethical concerns and limitations are inherent to most AI-based medical imaging technologies (and many of the general AI technologies too) today. On the other hand, the open discussion about these limitations that this paper has already elicited during the rebuttal period (and hopefully further at NeurIPS) would be of value to the ML community. I believe that the paper has a strong contribution being the first to propose to use NN for forensic dental identification. Given that neural networks are starting to be applied for forensic identification tasks, both the proposed approach and the clarifying discussion about limitations, ethics, and potential mis-use that the authors promised to add will be of great value to to this conference.

---

### Decision · Program_Chairs · 2021-09-27

**Decision:**

Accept (Poster)

**Comment:**

This paper generated significant debate, both on technical contribution considerations (more general methodology) and ethics considerations. Thank you to the reviewers and authors for the detailed discussion.

On the ethics considerations, the authors have responded saying they will include a more detailed discussion of potential misuses, and importantly, results across different subpopulations. The authors should include as many of these subgroups as possible (across gender, race and multiple age brackets) along with discussion on any places where the method performs below average.

On the technical considerations, a big debate was between whether the methods proposed are too specialized to this application. After careful review of the discussion, I think this paper can be viewed as an applied contribution, that should be of interest to others in the community who are studying this application.